# Consumer acceptance of new plant-breeding technologies: An application to the use of gene editing in fresh table grapes

Azhar Uddin[1], R. Karina Gallardo[2]*, Bradley Rickard[3], Julian Alston[4], Olena Sambucci[4]

1 Postdoctoral Research Associate, Institute for Research and Education to Advance Community Health (IREACH), Washington State University, Spokane, Washington, United States of America, 2 School of Economic Sciences, Puyallup Research and Extension Center, Washington State University, Puyallup, Washington, United States of America, 3 Charles H. Dyson School of Applied Economics and Management, Cornell University, Ithaca, New York, United States of America, 4 Department of Agricultural and Resource Economics, University of California, Davis, Davis, California, United States of America

* karina_gallardo@wsu.edu

**Data Availability Statement:** All relevant data are within the paper and its Supporting Information files.

## Abstract

This study estimates consumers' willingness to pay for specific product (quality) and process (agronomic) attributes of table grapes, including taste, texture, external appearance, and the expected number of chemical applications, and for the breeding technology used to develop the plant. Considering varietal traits, on average our survey respondents were willing to pay the highest price premiums for specific offers of improvements in table grape taste and texture, followed by external appearance and expected number of chemical applications. Considering breeding methods, on average our respondents were willing to pay a small premium for table grapes developed using conventional breeding rather than gene editing (e.g., CRISPR). Results from a latent class model identify four different groups of consumers with distinct preferences for grape quality attributes and breeding technologies. The group of consumers most likely to reject gene editing considers both genetic engineering and gene editing to be breeding technologies that produce foods that are morally unacceptable and not safe to eat.

## Introduction

Since genetically engineered crops were first introduced in the mid-1990s, they have faced considerable barriers to market acceptance. In these crops, genetic engineering tools are used to move genes from one non-closely related or sexually incompatible species to another [1,2]. The scientific consensus has been that the risks from genetically engineered crops to human health, society, and the environment are no greater than those for varieties produced using conventional breeding techniques [3,4]. Despite the scientific evidence, as discussed by Qaim [3,4] various groups (notably the environmental NGOs, Green Peace, and Friends of the Earth) have actively opposed the technology and publicized counterclaims, and some consumers perceive genetic engineering technologies and the foods produced from them as risky, unethical, or unnatural [5–7].

**Funding:** [RKG, BR, JA, AU, OS received funding This work is supported in part by the USDA National Institute of Food and Agriculture - Specialty Crop Research Initiative project "VitisGEN2" (2017-51181-26829) https://www.vitisgen2.org/ https://nifa.usda.gov/funding-opportunity/specialty-crop-research-initiative-scri The funders had no role in study design, data collection and analysis, decision to publish, or preparation of the manuscript.]

**Competing interests:** The authors have declared that no competing interest exist.

Limitations on their public and market acceptance have prevented genetic engineering technologies from realizing their full market potential, and concerns have been raised that other new breeding technologies, such as gene editing, may face similar barriers and suffer similar consequences [4,8]. Indeed, Lusk, Roosen, and Bieberstein [9] argue that opposition to genetically engineered foods has spilled over to affect the adoption of other breeding technologies in agriculture worldwide. Whether, to what extent, and over what time horizon this may occur is of concern to public agencies and private plant-breeding firms deciding which types of plant breeding technologies to employ, and are questions that came to the fore in the *Vitis-Gen2* project from which the current study was drawn.

This study contributes to advancing the knowledge about consumer acceptance of gene editing and foods produced using this technology. In our case study, based on data elicited using an online consumer survey, we specifically examine consumer demand for table grapes produced with and without gene editing (i.e., using conventional modern breeding methods). We consider the differential demand response when these alternative plant breeding methods are used to improve various fruit quality attributes and agronomic characteristics.

Table grapes make for an excellent case study, for three reasons. First, like many other perennial crops, table grape production relies heavily on the use of pesticides and gene editing could be used to develop table grapes that would require much less use of environmentally harmful crop protection products [10]. Second, over the past 30 years the table grape industry has enjoyed very rapid rates of development, commercialization, and adoption of new cultivars (Fig 1). These new table grape cultivars have been selected for both production-related reasons (such as yield, harvest timing, pest resistance, and resilience to extreme weather conditions) and traits (or attributes to improve) that are important to consumers (such as color, seedlessness, flavor, berry size and shape, and seasonal availability). Moving forward, breeding programs will continue to develop new cultivars to advance production attributes, improve traits

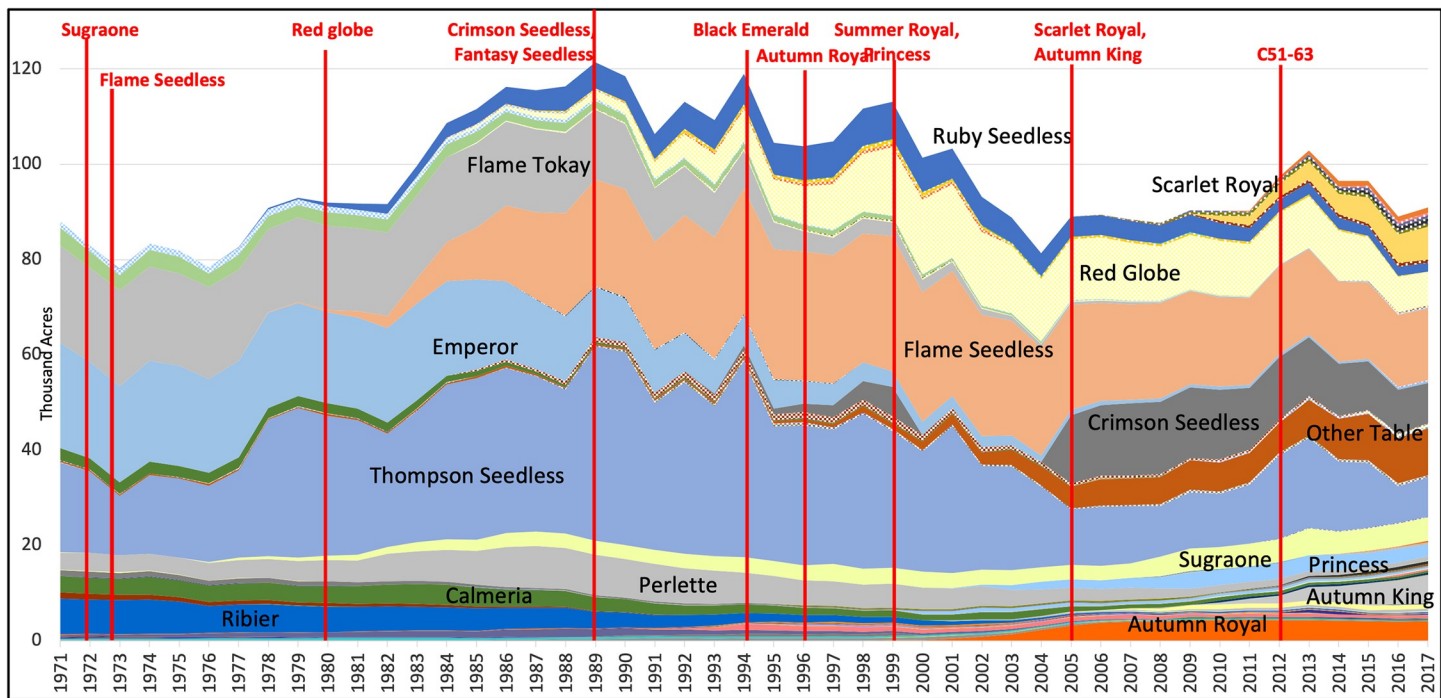

**Fig 1. Total bearing acreage by variety of table grapes in California, annual 1971–2017.** Source: Alston, Sambucci, and Serfas [Unpublished].

that are important to consumers, and to respond to various sustainability challenges including changes in climate and in environmental regulations on pesticide use [11]. But consumer acceptance may be a crucial consideration in deciding how best to accomplish those breeding objectives, and which objectives to emphasize.

Third, and lastly, much research has examined consumer demand for food products produced using gene-edited ingredients, but these studies have largely focused on annual crops [12]. The issues facing perennial crops (such as table grapes) are different for two reasons: i) these crops require a much larger investment per acre and a much longer planning horizon; and ii) the final consumption product often involves very little processing and consumers might be more concerned about breeding technologies used to produce fresh fruit and vegetable products [13]. Therefore, extending the limited previous work examining consumer acceptance of gene editing for perennial fruit crops (Shipman et al. [14]; Murigai et al. [15]; and Nagamangala Kanchiswamy et al. [16]), we hope to be able to shed new light on the tradeoffs surrounding gene editing in the context of consumer decisions about purchasing fruit. We expect that our results can be more widely applied to similar questions about consumer acceptance facing the other major fruit and vegetable breeding programs that are considering the adoption and commercialization of gene-editing techniques.

Our research makes two contributions to the growing literature on consumer acceptance of new plant-breeding technologies—in particular, gene editing. First, we assess the differences in consumers' willingness to pay (WTP) for a bundle of attributes of table grapes, including fruit taste and texture, external appearance, expected number of chemical applications and breeding technologies (gene editing versus conventional breeding). A generalized multinomial logit estimation is applied to data collected via a discrete choice experiment in a U.S. nationwide consumer survey. The study encompasses i) bundles of *product attributes* of table grapes that imply a direct and tangible benefit to consumers, and ii) bundles of *production process* attributes of table grapes that imply non-tangible benefits that accrue potentially to both consumers and producers, such as benefits from reductions in the expected number of chemical applications. Second, we identify consumer groups that differ in their WTP for different product attributes and breeding technologies. We apply a latent class model to identify potential sources of heterogeneity of preferences among groups of consumers. Here we examine the impact of sociodemographics, sources of trusted information and the level of knowledge of gene editing.

## Background

Gene editing enables scientists to manipulate the DNA by removing, inserting, or replacing portions of DNA of living organisms, with applications for bacteria, plants, and animals. The manipulated DNA can be induced by four systems including meganucleases, zinc-finger nucleases (ZFNs), transcription activator-like effector nucleases (TALENs) and clustered regularly interspaced short palindromic repeat systems (CRISPR-Cas systems, referred in this manuscript as CRISPR) [17–19]. Of these four forms of gene editing, CRISPR is simpler, faster, cheaper, and more accurate than other gene-editing technologies and hence many scientists performing gene editing now use CRISPR [20,21]. By 2020, the use of CRISPR by plant breeding programs was rapidly expanding, as more and more plants with market-oriented traits were being developed, and companies had already released CRISPR crops to the market [22].

CRISPR has been applied to diverse agricultural crops including camelina sativa (oil seed plant), casava, flaxseed, grapefruit, maize, mushrooms, orange, potato, rice, soybean, tomato, and wheat [19]. Other applications of CRISPR include peanuts, kiwi, lettuce, lemon, poppy, salvia, cacao, banana, manioc, and sugar cane [22]. Target traits using CRISPR are agronomic,

food and feed quality, and tolerance of biotic and abiotic stresses. Note that the abundant literature on the scientific investigations and application of CRISPR to different crops does not offer evidence of strategic decisions for further development or market releases of the resulting technologies.

While this study focuses on application of CRISPR to plant crops, we build on previous studies of applications of gene editing to animals. CRISPR system technologies have been used to treat genetic disorders in animals, expedite livestock breeding, and to engineer new antimicrobials and control disease-carrying insects [23]. If successfully applied, gene editing can improve productive traits and disease resistance in animals, making livestock production more efficient and environmentally sustainable [24]. Examples of applications include Superfine Merino sheep that produce the highest quality of wool and improved meat production, and pigs resistant to the Porcine Respiratory and Reproductive Syndrome virus. Gene editing also has potential to improve animal welfare, for example, gene editing can be used to avoid horn removal in calves [24].

Government regulations of gene-edited organisms vary widely. The European Union requires the same environmental, food and feed risk assessments for gene-edited crops as for genetically engeneered crops. Meanwhile, the United States, South American countries (Argentina, Brazil, Chile, and Colombia), Australia, and Japan exempt gene-edited organisms from regulation, as long as these organisms do not contain novel combinations of genetic materials that otherwise could not be developed through conventional breeding [25]. In the United States, gene-edited plants are already being marketed. By 2021, gene-edited Calyno oil was commercialized in the United States. This oil resulted from a highly oleic soybean and contains less saturated fat and lower oil absorption compared to conventional soybean oil. Also, at that time gene-edited Sulfonylurea-tolerant (SU) canola was available in the United States and Canada [25]. In March 2022, the U.S. Food and Drug Administration (FDA) announced regulatory clearance for gene-edited beef cattle [26].

In light of the increasing likelihood that the number of applications will continue to grow, and over the next decade many gene-edited agricultural crops will be commercialized, it is timely to assess the perceptions of consumers towards this new plant-breeding technology. It is important specifically to determine where consumers will take a negative view of, or even show some trepidation regarding, products from crops using these new plant-breeding technologies as was witnessed with genetically engineered crops.

## Consumer response to food produced with gene editing

Our research is motivated by questions about the extent to which consumers can be expected to view gene editing as yet-another iteration of genetic distortions of nature with unpredictable health or environmental consequences—as some see genetic engineering technologies in agriculture, contrary to the positions taken by scientific authorities and American regulators [25].

Hall (2016) reports that, compared with genetic engineering, some scientists are optimistic that consumers will be more receptive to foods produced using gene editing, as gene-edited crops are "fundamentally different" from genetically engineered crops [27]. Unlike genetic-engineering technology that was developed by industry, gene editing, specifically CRISPR was developed by academia, and scientists developing and applying CRISPR are making all information about the technology and its applications public. In addition, CRISPR is more affordable than genetic engineering, making it accessible to a wider variety of institutions and companies [28–30]. However, early evidence shows that i) like genetically engineered crops, compared to conventionally bred crops gene-edited crops are viewed by some consumers as providing inferior food products, and ii) some organizations that have historically opposed

genetic engineering will lump gene editing together with genetic engineering, opposing both [31].

Findings from meta-analyses of numerous studies, indicate that, in general, across different regions of the planet, consumers are willing to pay premiums to avoid genetically engineered foods [32–34]. The extent of consumer aversion to genetically engineered foods varies among products and across consumers, and this variation depends largely on the direct benefit perceived by consumers, their trust in government and regulatory agencies, and their level of knowledge of each breeding technique. As noted by the U.S. Department of Agriculture, Economic Research Service [35], the predominant genetically engineered crops grown widely in the United States (and in the world more generally) are cotton, soybeans, canola, and corn, for which the improved traits are mainly agronomic (namely increased yields, herbicide tolerance, and insect resistance). Hence, while they have benefited consumers considerably by making food more abundant and cheaper, the genetically engineered crops that have been adopted to date are largely not perceived by consumers to directly contribute to their wellbeing or to lead to outcomes that they consider to be important. There are some exceptions, e.g., Arctic$^{TM}$ apples that are genetically engineered to slow down flesh discoloration (the non-browning trait) that is directly observed by consumers, and has the potential to reduce food waste [13,36]. Wunderlich and Gatto [37] conclude that in general, U.S. consumers' knowledge of genetically engineered food is low, and that awareness does not increase with the frequency of purchases. Examples of commercially unsuccessful genetically engineered crop varieties include Flavour Saver tomato released in 1994 and the NewLeaf$^{TM}$ potato that was resistant to the Colorado potato beetle was relased in the late 1990s but was subsequently withdrawn in 2001 [38].

Applications of gene editing go beyond improvements in agricultural crops; they can be applied to any living organism, including animals and humans [21]. Results from different studies show that, in general, the use of gene editing in humans and animals is of concern to some individuals, notwithstanding a very broad scientific consensus as to its safety for the environment and human health [21,39,40]. For example, Critchley et al. [21] showed that individuals could be more supportive if the result is enhanced human health, but no support is shown for applications to animals for human food or to enhancement of human appearance. Watanabe et al. [39] underscore the increasing controversy surrounding the application of gene editing to humans, either for therapeutic or function enhancement. Funk et al. [40] showed that Americans are concerned about using gene editing to reduce the risk of diseases in humans.

With respect to plant crops, an increasing number of studies in the agricultural economics literature have analyzed consumers' preferences and their WTP for gene-edited crops: rice [12], apples [7,36], and potatoes [15]. These studies found that consumers were more receptive of gene editing compared to genetic engineering, but that gene editing is less accepted than conventional breeding. Findings were consistent among countries, albeit with differences in the magnitude of the discounts for gene-edited products; for example Shew et al. [12] found a larger discount in Europe compared to the United States, Canada, and Australia for both gene-edited and genetically engineered rice compared with conventionally bred rice. Marette, Disdier, and Beghin [36] found that the price discount for gene-edited and genetically engineered apples is smaller in the United States than in France. Muringai, Fan, and Goddard [15] found that the discount varied with the nature of the improvement. Participants in their survey required a smaller discount for gene-edited potatoes with health-related improvements (lowering acrylamide, a component released when potatoes are fried) compared to environmentally related improvements (pesticide and food waste reduction).

Some studies have explored consumers' WTP for gene-edited meat products [41] and found that in general consumers would only accept gene-edited meats at discounted prices. The discount decreased when consumers were informed that the benefit from gene editing was improved animal welfare associated with increased disease resistance or consequences for human health through increased Omega 3 fatty acids [41].

Our work extends research in this arena by analyzing preferences for different bundles of attributes for table grapes while taking the breeding technology into account. This enables us to identify whether it is the tangible benefits (such as improvements in taste and texture or external appearance) or the non-tangible benefits (such as number of chemical applications) that affect consumer acceptance of gene editing in table grape production.

## Data

Our study developed a series of choice experiments to collect information about how consumers value specific bundles of attributes of table grapes, including cultivar-related attributes of both the product, such as grape taste, texture, and external appearance, and the process used to produce it, such as the expected number of chemical applications. The data were collected online via the survey platform Qualtrics during April 2020. After the inconsistent responses were removed, the survey included responses from a total of 2,873 participants, comprised of subjects that: i) collectively were broadly consistent with a random representation of U.S. demographics in terms of their age and geographical location, ii) individually were the primary grocery shoppers in their respective households, and iii) had consumed table grapes during the previous three months. The survey tool was approved by the Washington State University Institutional Review Board (IRB) for use with human subjects. The IRB number is 18186–001.

To examine respondents' preferences for bundles of product and process attributes of table grapes, including varietal traits, coupled with their preferences for breeding technologies, we conducted a discrete choice experiment. The survey contained an explanation of the discrete choice experiment, along with a description of each attribute included. That is, respondents were provided with a detailed definition of both conventional breeding and gene editing. Because we explicitly indicated in the survey that gene editing referred to CRISPR, we will use CRISPR to represent gene editing in the description of the study and discussion of results (Fig 2).

Each respondent was presented with eight scenarios, one by one, each of which was designed to mimic a grocery store shopping experience for table grapes. Each scenario contained two purchase options, randomly assigned as option A and option B, each of which presented table grapes having a specific combination of five (bundles of) attributes: i) flavor and texture, ii) external appearance, iii) expected number of chemical applications, iv) breeding method, and v) price. The specific combinations of attributes in each option, A and B, are chosen and combined at random from the alternatives displayed in Table 1, which presents the list of attributes and the alternative possibilities available for each. In each scenario, consumers were asked to select only one option; they could choose to buy option A or option B, or neither option A nor B (which was labelled as option C in each scenario). An example of a choice scenario is presented in Fig 3.

All possible combinations of attribute levels or a full factorial design would have yielded 32 different scenarios. Asking each subject to complete 32 choice experiments would be expected to create respondent fatigue and would compromise the reliability of the study. Hence, we used the JMP® software to generate a fractional factorial design. This software employed a two-step procedure following Kessels, Jones, and Goos [42] and selected eight choices, ensuring orthogonality, balance, and maximizing the D-efficiency. An orthogonal design implies

Each table grape variety was developed using one of the following breeding techniques:

- **Conventional breeding:** Plants with desirable traits are bred together, using existing varieties or the offspring of previous breeding programs that have the desired traits. This results in hundreds of potentially desirable plants that must be whittled down to the best candidates for commercial use. May be labelled as organic (if other production requirements are satisfied) or GMO-free.

- **Gene editing (e.g. CRISPR):** Specific genes can be altered, without introducing genes from any other sources. Similar to editing a word in a novel, gene editing can target specific DNA sequences in the genome for slight modification, which can improve plant traits. The USDA recently proposed that plants produced using gene editing will be treated the same as conventionally bred plants. For this study we can assume grapes produced using gene-editing may be labeled as organic (if other production requirements are satisfied) or GMO-free.

**Fig 2. Definitions describing conventional breeding and gene editing (CRISPR) provided to survey respondents prior to the discrete choice experiment scenarios.** Scenario 1. Of the two purchase options below, choose the ONE you would most likely buy. If neither is acceptable, please choose Option C–i.e., neither Option A nor Option B.

that all estimable effects are uncorrelated, while a balanced design ensures that each level appears equally often within each attribute. The D-efficiency is a measure of the goodness of a design relative to an hypothetical orthogonal desing. These measures are based on the variance-covariance matrix of the vector of parameter estimates. An efficient design is the one with a small variance matrix with the eigenvalues of the inverse matrix providing measures of the design size. D-efficiency is a function of the geometric mean of the eigenvalues [43].

**Table 1. List of attributes and attribute levels used in the choice experiment.**

| Table grape attributes | Alternative possibilities available for each attribute | |
|---|---|---|
| Fruit taste and texture<br> Sweetness, flavor, crispness, and firmness. | Combination of weak/mild flavor and mealy/not firm texture | Combination of strong/full flavor and crisp/firm texture. |
| External appearance<br> Presence of defects, fruit size, and green color uniformity. | Excellent | Poor |
| Expected number of chemical applications<br> Pesticides for insect pests and fungicides for fungal diseases. | Current number of applications | 80% lower than current number of applications |
| Breeding technique<br> Identifying and selecting desirable traits in plants and combining into one individual plant. | Conventional breeding | Gene editing (CRISPR) |
| Price | $1.98/lb ($4.37/kg) | $2.98/lb ($6.57/kg) |

**SCENARIO 1.** Of the two option below, choose the ONE you would most likely buy. If none of the choices are acceptable, please choose Option C: Neither option A nor B.

| Attributes | Option A | Option B | **Option C** |
|---|---|---|---|
| **Fruit taste and texture** <br> *E.g., sweetness, flavor, crispness, and firmness.* | Inferior <br> (Combination of weak/mild flavor and mealy/not firm texture) | Superior <br> (Combination of strong/full flavor and crisp/firm texture) | Neither option A nor B |
| **External appearance** <br> *E.g., presence of defects, fruit size, and green color uniformity* | Excellent | Poor | |
| **Expected number of chemical applications** <br> *E.g., pesticides for insect pests and fungicides for fungal diseases.* | Same as current number of applications | Same as current number of applications | |
| **Breeding technique** <br> *Identifying and selecting desirable traits in plants and combining into one individual plant.* | Gene editing | Gene editing | |
| Price ($/lb) | 2.98 | 2.98 | |

I would choose Option A I would choose Option B I would choose Option C

○ ○ ○

**Fig 3. Example of a choice experiment scenario.**

In addition to the discrete choice questions, respondents were asked several questions about their preferences for table grape attributes, their table grape consumption, and their perceptions about science and technology. Finally, they were asked a series of questions about their sociodemographic details. The full set of survey questions is available upon request from the authors.

## Empirical approach

The empirical approach of this paper is based on Lancaster [44] and McFadden [45]. First, following Lancaster's [44] theory of demand for characteristics, we assume that consumers derive utility from the attributes inherent in the good rather than from the good itself. From McFadden [45] we follow random utility theory and model the utility of the consumer as being composed of a deterministic component, given by the good's attributes, and a random component, given by unobserved factors.

To compute the parameter estimates, we use the Generalized Multinomial Logit Model (G-MNL) [46]. The G-MNL allows for "scale" heterogeneity, which implies that, given the

attribute coefficients are assumed to be fixed, the scale of the idiosyncratic error term is greater for some consumers than for others. This means that choice behavior is more random for some consumers compared to others as the scale of the error term is inversely related to the error variance [47]. Accordingly, the utility of respondent $n$ choosing alternative $j$ in choice scenario $t$ is

$$U_{njt} = \beta_{0j} + [\sigma_n \beta + \gamma \eta_n + (1 - \gamma)\sigma_n \eta_n] x_{njt} + \lambda p_{njt} + \varepsilon_{njt} \tag{1}$$

where $\beta_{0j}$ is the alternative specific constant for each alternative $j$, $\sigma_n$ is a random variable that captures the scale heterogeneity, $\beta$ is a constant vector, $\gamma$ is a scale parameter, $\eta_n$ is a vector of indiidual specific taste deviations from the mean based on observed attributes that captures residual preference heterogeneity, $x_{njt}$ is the vector of the observed attributes of choice, and $\varepsilon_{njt}$ is an unobserved error term. The random variable $\sigma_n$ follows,

$$\sigma_n = \exp(\bar{\sigma} + \theta z_n + \tau v_n) \tag{2}$$

where $\bar{\sigma}$ is the mean, $z_n$ is a vector of characteristics associated with individual $n$, $\tau$ is the standard deviation, and $v_n$ follows a standard normal distribution $(0,1)$; $\bar{\sigma}$ is a normalizing constant such that $\sigma_n$ is equal to 1 when $\theta = 0$.

This study reports the results from estimation using the Type I or GMNL-I version of the model that assumes $\gamma = 1$, or that the standard deviation of the residual taste heterogeneity is proportional to the scale parameter. We also estimated the Type II or GMNL-II version that assumes $\gamma = 0$, or that the standard deviation of residual heterogeneity is independent of the scale parameter. Goodness of fit statistics (the Akaike Information Criterion (AIC), the Bayesian Information Criterion (BIC) and the likelihood function) indicate that the GMNL-I model outperforms the GMNL-II (with a lower AIC and BIC). Therefore the GMNL-I estimates are reported.

A latent class analysis was subsequently performed to assess if the WTP for table grapes developed using different breeding techniques varied across groups of consumers that share common unobservable characteristics. We did this to gauge the impact of reactions to different sources of information, levels of knowledge and perceptions of plant breeding, and general socioeconomic characteristics.

## Latent class model

The latent class model captures heterogeneous preferences across consumers and identifies classes (hereafter we will refer to groups instead of classes) within the sample of survey respondents. Accordingly, individuals can be sorted into a number of latent or unobservable subgroups based on their responses to other questions in the survey; these other responses are the membership function variables. Preferences across groups are heterogeneous, but preferences among individuals within each group are assumed to be homogeneous [48,49]. Mathematically, the probability that individual $n$ will choose alternative $i$ in choice scenario $j$ for latent group $c$ is:

$$Pr(nij|c) = \frac{\prod_{j=1}^{J} e^{\beta_c x_{nij}}}{\sum_{i=1}^{I} e^{\beta_c x_{nij}}}, \tag{3}$$

where $x_{nij}$ is the vector of observed attributes associated with alternative $i$, $\beta_c$ is the estimated vector of group-specific utility parameters, which captures preference heterogeneity among groups, and $j$ indicates the set of choice scenarios available to individual $n$. A fractional

multinomial logit model is used to estimate the probability that individual $n$ belongs to group $c$:

$$Pr(c) = \frac{e^{\theta_c m_n}}{1 + \sum_{c=1}^{C-1} e^{\theta_c m_n}},\tag{4}$$

where $m_n$ is the set of observable individual characteristics that affects the group membership vector $\theta_c$, (the $c^{th}$ parameter vector is normalized to zero to ensure identification of the model). The model estimates the probability of a specific choice for individual $n$ as the expected value, over groups, of the group-specific probabilities. In our choice experiment, each respondent was asked to make choices for eight different scenarios. The observation of repeated choices by the respondents helps us to examine how levels of various attributes affect individual utility and to compare choice responses across different classes or groups of respondents [47].

To identify the number of groups (or latent classes) practitioners use a set of fit indices. These include measures of goodness of fit, interpretability of results, and classification diagnosis. The latter enables the identification of how well the groups are classified and differentiated within the sample of survey respondents [49]. The measures of goodness of fit are presented in Table 2. One can observe that increasing the number of groups led to an increased likelihood function, and decreased Akaike Information Criterion (AIC) and Bayesian Information Criterion (BIC); which is preferred. However, one can observe "diminishing returns" after three groups are defined, and the prediction accuracy decreases as more groups are added. In this study, besides the goodness of fit measures, the following criteria are used [49]: (1) how the selected models relate to each other, for example to identify if one model is an expanded version of the other and (2) the relative sizes of the groups, which are dependent on the total number of responses. In this paper we opted for four groups or latent classes based on the statistical significance and magnitude of parameter estimates in each group showing that there is no group that is an expanded version of the other, and the measures of goodness of fit.

## Summary statistics and empirical results

### Sociodemographic characteristics of respondents

Table 3 presents summary statistics describing the sociodemographic characteristics of the survey respondents, and compares them with the corresponding information in the U.S. Census [50]. Fifty-nine percent of the respondents in the sample are female, 75% are of white ethnicity, 53% have at least a 4-year degree, 48% have children under 18 years old in the household, 26% have worked or lived on a farm or ranch, and 17% are vegetarian. The average age of the respondents is 42 years, the average yearly household income is $99,922, and the average household size is three individuals. Compared with the 2018 U.S. Census averages, our sample includes a greater share of females, a larger proportion of individuals with at least a 4-year college degree, and on average, individuals with a higher income [50]. However, our survey respondents follow the profile of individuals who tend to be more responsive to surveys [51].

**Table 2. Goodness of fit criteria to select the number of groups in the latent class model.**

| Number of groups or latent classes | Parameters | Likelihood function | AIC | BIC | Prediction accuracy |
|---|---|---|---|---|---|
| 2 | 35 | -22189.91 | 44449.81 | 44658.52 | 0.954 |
| 3 | 64 | -21604.59 | 43337.19 | 43718.83 | 0.949 |
| 4 | 93 | -21265.64 | 42717.28 | 43271.85 | 0.893 |
| 5 | 122 | -20943.54 | 42131.08 | 42858.58 | 0.894 |

**Table 3. Demographic characteristics of survey respondents compared to U.S. Census, categorical variables.**

| Item | Survey sample (N = 2,873) | U.S. Census 2018[a] |
|---|---|---|
| | *Percentage of respondents* | |
| Female | 58.7 | 50.8 |
| Race | | |
| White/Caucasian, European American | 75.4 | 75.5 |
| Asian, Asian American | 8.2 | 5.4 |
| Black, African American | 7.7 | 14.0 |
| Hispanic or Latino American | 6.5 | 17.8 |
| American Indian or Alaskan Native | 1.0 | 1.7 |
| Middle Eastern, Middle Eastern American | 0.6 | – |
| Pacific Islander | 0.2 | 0.4 |
| Other (Human, Mixed, Spain, Greek etc.) | 0.5 | – |
| Education | | |
| 4-year degree | 29.5 | 19.4 |
| Postgraduate degree | 23.7 | 12.1 |
| Some college | 15.5 | 20.6 |
| High school graduate | 15.4 | 27.1 |
| 2-year degree | 9.1 | 4.2 |
| Professional degree | 5.6 | 4.2 |
| Less than high school | 1.3 | 12.4 |
| Other (Certificate, dropped out) | 0.1 | – |
| Income | | |
| Less than $25,000 | 8.6 | 20.2 |
| $25,000–$34,999 | 6.6 | 9.3 |
| $35,000–$49,999 | 5.7 | 12.6 |
| $50,000–$74,999 | 19.1 | 17.5 |
| $75,000–$99,999 | 15.1 | 12.5 |
| $100,000–$149,999 | 21.1 | 14.6 |
| $150,000–$199,999 | 10.0 | 6.3 |
| More than $200,000 | 9.4 | 7.0 |
| Prefer not to answer | 4.3 | – |
| Percent of households with children under 18 | 47.7 | 41.5 |
| Worked/Lived in a farm or ranch | 26.0 | – |
| Vegetarian | 16.9 | – |
| | *Average* | |
| Age | 42.4 | 38.2 |
| | (15.9)[b] | |
| Annual household income | 99,922.2 | 63,179 |
| | (61,542.6) | |
| Number of members in the household | 2.9 | 2.6 |
| | (1.3) | |

[a]Source: US Census Bureau [41].

[b]Standard deviations are shown in parentheses.

**Table 4. Frequency distribution of respondents describing table grape consumption features.**

| Table grape consumption features | Percentage of responses in each category (N = 2,873) |
|---|---|
| Consumption frequency | |
| Every 2–3 weeks | 35.64 |
| Every 2–3 months | 21.27 |
| 1–2 times per week | 20.89 |
| 3–4 times per week | 8.08 |
| 2–3 times per year | 7.31 |
| 4 or more times per week | 5.43 |
| Less than 2 times per year | 2.09 |
| Reason for not consuming more often-less than 2 times per year | |
| Have a preference for other fruit | 32.12 |
| Too expensive | 23.04 |
| Availability/access to table grapes | 19.30 |
| Other (spoil fast, variety etc.) | 9.31 |
| Don't like the flavor | 7.72 |
| Don't like the external appearance | 3.18 |
| Don't like the texture | 2.95 |
| Preparation time (i.e., washing) | 2.38 |
| Preferred table grape package | |
| Pre-bagged | 58.65 |
| Clamshell | 23.77 |
| Loose | 17.58 |
| Type of table grape often bought | |
| Green | 47.86 |
| Red | 42.12 |
| Black | 8.15 |
| Other (Mix, unsure etc.) | 1.88 |

We also present responses for questions used to elicit information on shopping and eating habits such as ratings of importance of table grape quality attributes, ratings of importance for label information, ratings of importance for trusted sources of information, and perceptions of plant breeding methods. Table 4 presents the frequency distribution for responses to questions about respondents' shopping and eating habits. On average, 36% of the respondents in the sample consume table grapes every 2–3 weeks, 21% consume table grapes every 2–3 months, and 20% consume table grapes 1–2 times per week. Of the respondents (2.09% of the total sample) who consume table grapes fewer than two times per year, 32% indicated they prefer other fruit; 23% indicated table grapes are too expensive, and 19% indicated they do not consume more because they are not available. On average, 59% of the respondents indicated they prefer pre-bagged table grapes while 48% indicated that their favorite type of table grape is green.

Table 5 presents the ratings of importance for a list of table grape attributes; a 1–5 scale was used where 1 = very unimportant and 5 = very important. The list was divided into three sections: i) taste and texture, ii) appearance, and iii) other attributes. The "taste and texture" section includes freshness, ripeness, juiciness, firmness, sweetness, crispness, thickness of berry skin, aroma, tartness, and unique flavor. The "appearance" attributes include berries that appear to be free from defects, a uniform and attractive berry color, stems appear green rather than dried out, specific fruit size, and uniform size and shape. The "other attributes" section

**Table 5. Ratings of importance assigned to table grape attributes.**

| Table grape attributes | Mean[a]<br>1 = very unimportant, 5 = very important |
|---|---|
| Taste- and texture-related | |
| Freshness | 4.53 |
| | (0.84) |
| Ripeness | 4.33 |
| | (0.89) |
| Juiciness | 4.31 |
| | (0.88) |
| Firmness | 4.23 |
| | (0.90) |
| Sweetness | 4.18 |
| | (0.89) |
| Crispness | 4.56 |
| | (0.95) |
| Thickness of berry skin | 3.58 |
| | (1.00) |
| Aroma | 3.50 |
| | (1.08) |
| Tartness (acidity) | 3.45 |
| | (1.06) |
| Unique flavor (e.g., cotton candy) | 3.32 |
| | (1.29) |
| Appearance-related | |
| Berries appear free from defects (brown spots, cracks, etc.) | 4.27 |
| | (0.97) |
| Uniform and attractive berry color | 3.90 |
| | (1.02) |
| Stems appear green rather than dried out | 3.83 |
| | (1.03) |
| Specific fruit size (large, medium, small berries) | 3.71 |
| | (1.00) |
| Berries are of uniform size and shape | 3.61 |
| | (1.06) |
| Other | |
| Seedlessness | 4.16 |
| | (1.03) |
| Phytonutrient content (e.g., vitamins, antioxidants) | 3.79 |
| | (1.04) |

[a] Standard deviations are shown in parentheses.

includes seedlessness and phytonutrient content. On average, a higher rating was assigned to taste and texture related attributes (3.96) compared to appearance-related attributes (3.86).

Table 6 presents ratings of importance for different pieces of information displayed on the labels; a 1–5 scale was used where 1 = very unimportant and 5 = very important. We have four types of label information: i) chemical application, ii) breeding technology, iii) origin, and iv) other label information. The "chemical application" labels include pesticide-free, sustainable agriculture, eco-label, and organic. The "breeding technology" labels include non-genetically

**Table 6. Respondents' ratings of importance of different food labels.**

| Food label | Mean[a]<br>1 = very unimportant, 5 = very important |
|---|---|
| Chemical application related | |
| Pesticide free | 3.81 |
| | (1.32) |
| Sustainable agriculture | 3.36 |
| | (1.37) |
| Eco-label | 3.36 |
| | (1.42) |
| Organic | 3.27 |
| | (1.41) |
| Breeding technology related | |
| Non-GMO | 3.44 |
| | (1.43) |
| Name of the grape variety | 3.16 |
| | (1.36) |
| Origin | |
| Domestic product | 3.52 |
| | (1.24) |
| Local origin | 3.32 |
| | (1.33) |
| Seedlessness | 4.05 |
| | (1.17) |
| A private brand | 2.82 |
| | (1.62) |

[a] Standard deviations are shown in parentheses.

engineered and the name of the grape variety. The "origin" labels include domestic production and local production. "Other label information" includes seedlessness and private brand. On average, the highest rating was assigned to seedlessness (4.05). Chemical application attributes were assigned a higher average rating (3.31) compared to breeding technology attributes (3.17). The attribute of private brand was given the lowest rating of importance (2.82).

Table 7 presents the ratings of importance for sources of information when making food purchase decisions; a 1–5 scale was used where 1 = strongly do not trust, and 5 = strongly trust. The most trusted sources of information for consumers are "scientific groups" (3.88), followed by "universities" (3.72), "producer-oriented groups" (3.70), "government" (3.51), and "consumer-oriented groups" (3.48). The least trusted source of information was "social media and media" (3.30). Similar results were found in a nationwide U.S. survey conducted by the PEW Research Center, with respondents selecting medical professionals and scientists as the information sources most likely to act in the public's best interests over military, police officers, public school principals, and religious leaders. Respondents expressed the lowest levels of confidence in journalists, business leaders, and elected officials [52].

Table 8 presents summary statistics for the questions about crop breeding and agricultural production methods. Fifty-eight percent of the respondents agreed with the statement that there was a difference between CRISPR and genetic engineering, but they did not know what it was, whereas 27% of the respondents agreed with the statement that there was a difference, and they knew what it was. Fifteen percent of the respondents indicated that there was no

**Table 7. Respondents' ratings of importance assigned to the trustworthiness of sources of information.**

| Source of information | Mean[a] 1 = very unimportant, 5 = very important |
|---|---|
| Scientific groups[b] | 3.88 |
| | (0.80) |
| Universities | 3.72 |
| | (0.98) |
| Producer-oriented groups[c] | 3.70 |
| | (0.78) |
| Government[d] | 3.51 |
| | (0.92) |
| Consumer-oriented groups[e] | 3.48 |
| | (0.89) |
| Social media, media[f] | 3.30 |
| | (0.87) |

[a] Standard deviations in parentheses.

[b] Scientific groups include medical professionals (e.g., your primary physician), scientific associations (e.g., American Association for the Advancement of Science), scientific journals (e.g., Nature, National Geographic).

[c] Producer-oriented groups include individual farmers, farmer's organizations (e.g., California Table Grape Commission), food manufacturers (e.g., Nestle, General Mills), food retailers (e.g., Walmart, Safeway).

[d] Government includes local government (e.g., local mayor) and government agencies (e.g., U.S. Department of Agriculture).

[e] Consumer-oriented groups include activist groups (e.g., Green America), consumer organizations (e.g., American Council of Consumers).

[f] Social, media, family and friends includes newspaper, TV, magazines, friends, and family members.

difference between the two breeding technologies. More than a half of the respondents to this survey perceived CRISPR as being different from genetic engineering although they did not know exactly what the difference was. In general, respondents to our survey indicated they are less informed about CRISPR compared with conventional breeding and genetic engineering. On the level of knowledge, Wunderlich and Gatto [37] conclude that in general, U.S. consumers' knowledge of genetically engineered food is low, but that compared with Italian and Japanese consumers, U.S. consumers were more likely to be at least somewhat familiar with genetically engineered foods.

Table 8 also includes results on perceptions about how safe, risky, natural, and morally acceptable are a list of production methods (conventional and organic farming methods) and a list of breeding technologies (conventional, genetic engineering, and CRISPR). Consumers rated products from organic farming as the safest to eat, the most natural, and the most morally acceptable, followed by conventional farming. Across the breeding technologies, consumers rated products from varieties developed by conventional breeding as the safest to eat, the most natural and morally acceptable, followed by those from CRISPR, and then genetic engineering. These results are aligned with findings by Shew et al. [12], Yan and Hobbs [7], Muringai, Fan, and Goddard [15], and Marette, Disdier, and Beghin [36], in that consumers have a more positive perception of gene-edited compared with genetically engineered foods.

We also asked respondents to rate their willingness to buy various food products across a range of breeding techniques (conventional breeding, CRISPR, and genetic engineering) on a 1–5 scale where 1 = least willing to buy and 5 = most willing to buy. The list of food products included fresh table grapes, fresh milk and raw potatoes, and also processed versions of each product (namely grape juice, ice cream, and french fries).

**Table 8. Respondents' ratings of importance assigned to perceptions on breeding methods.**

| Statement | % of respondents |
|---|---|
| Difference between "Gene editing (e.g., CRISPR)" and "Genetic engineering" | |
| Yes, there is a difference, but I don't know what it is | 57.81 |
| Yes, there is a difference and I know what it is | 27.12 |
| No, there is no difference | 15.07 |
| How informed respondents are on breeding methods | Average rating[a] (1 = completely uninformed, 5 = completely informed) |
| Genetic engineering | 3.25 |
| | (1.24) |
| Conventional breeding | 3.22 |
| | (1.23) |
| CRISPR | 3.02 |
| | (1.26) |
| Level of risk perceived (1 = highly risky to eat, 5 = totally safe to eat) | |
| Organic farming | 4.30 |
| | (0.86) |
| Conventional farming | 4.10 |
| | (0.87) |
| Conventional breeding | 3.84 |
| | (1.01) |
| CRISPR | 3.30 |
| | (1.11) |
| Genetic engineering | 3.21 |
| | (1.22) |
| How natural the methods are (1 = highly unnatural, 5 = completely natural) | |
| Organic farming | 4.27 |
| | (0.87) |
| Conventional farming | 4.05 |
| | (0.92) |
| Conventional breeding | 3.73 |
| | (1.06) |
| CRISPR | 2.88 |
| | (1.24) |
| Genetic engineering | 2.74 |
| | (1.30) |
| How ethical or morally acceptable are the following methods (1 = completely unethical, 5 = completely ethical) | |
| Organic farming | 4.32 |
| | (0.89) |
| Conventional farming | 4.12 |
| | (0.94) |
| Conventional breeding | 3.79 |
| | (1.10) |
| CRISPR | 3.23 |
| | (1.17) |
| Genetic engineering | 3.11 |

(*Continued*)

**Table 8.** (Continued)

| Statement | % of respondents |
|---|---|
| | (1.26) |

[a] Standard deviations are shown in parentheses.

Results from this question are presented in Fig 4, and here we see that the rating for the foods from conventionally bred varieties was the highest regardless of whether it was fresh or processed. The second-highest average rating was for the foods produced from varieties developed using CRISPR, and the lowest was for foods produced from varieties developed using genetic engineering. The difference between the ratings for foods from conventionally bred and gene-edited varieties is on average 0.72, while the difference between the rating for foods from gene-edited and genetically engineered varieties is 0.09 (on a scale between 1 and 5). These results reveal that although respondents perceive a difference between CRISPR and genetic engineering, this difference is not substantial. Lusk, McFadden, and Rickard [13] found that fresh foods received the highest discounts for being genetically engineered, but we did not find large differences between fresh and processed foods in the responses to our survey question.

## Willingness to pay results

Table 9 presents the WTP results from the GMNL-I model; recall, these results are applicable to the respondents to the survey in this study. Here, the estimated coefficients can be interpreted directly as estimates of the premium that consumers are willing to pay for each of the bundles of attributes as defined, relative to the default bundles. These results show that respondents are willing to pay the highest price premium, $1.44/lb ($3.17/kg), for improvements in

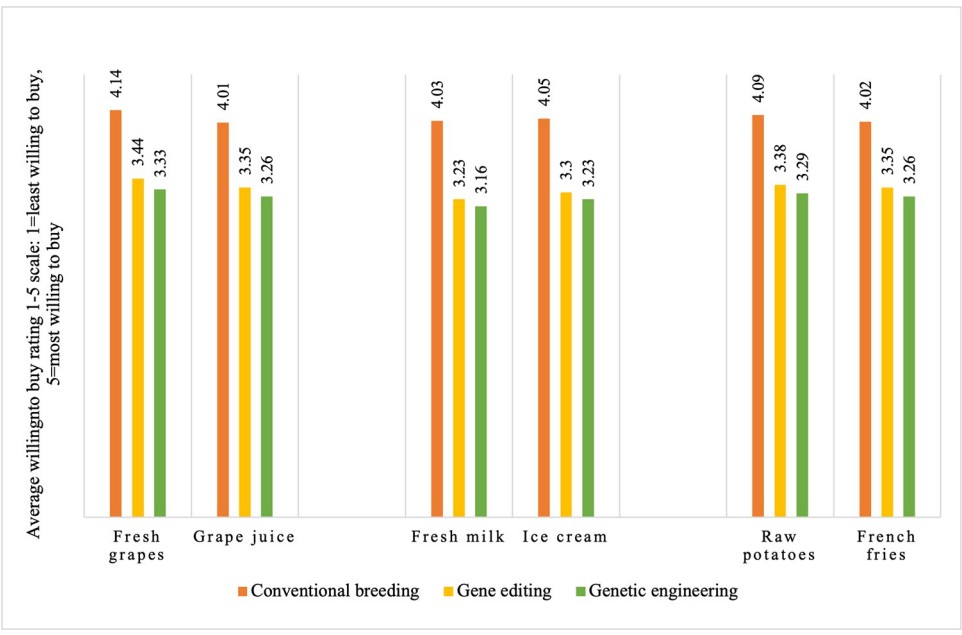

**Fig 4. Average rating for the willingness to buy on a scale of 1–5 where 1 = least willing to buy and 5 = most willing to buy.**

**Table 9. Coefficient estimates for the GMNL-I model, including selected table grape quality attributes and two breeding methods.**

| Variables | Coefficient estimates[a,b] | |
| --- | --- | --- |
| | **Mean** | **Standard deviation** |
| Price | -0.12*** | |
| | (0.03) | |
| Fruit taste and texture | 1.44*** | 0.31*** |
| e.g., sweetness, flavor, crispness, and firmness (combination of strong/full flavor and crisp/firm texture versus combination of weak/mild flavor and mealy/not firm texture) | (0.09) | (0.10) |
| External appearance | 0.86*** | 0.64*** |
| e.g., presence of defects, fruit size, and green color uniformity (excellent versus poor) | (0.06) | (0.04) |
| Expected number of chemical applications | 0.20*** | 0.11 |
| e.g., pesticides for insect pests and fungicides for fungal diseases (80% lower than current number of applications versus same as current number of applications) | (0.04) | (0.14) |
| Breeding technique | -0.36*** | 0.85*** |
| Identifying and selecting desirable traits in plants and combining into one individual plant (CRISPR versus conventional breeding) | (0.05) | (0.04) |
| Alternative specific constant—No purchase option | | |
| | 1.16*** | 1.62*** |
| | (0.12) | (0.05) |
| Scale heterogeneity parameter $\tau$ | 2.05*** | |
| | (0.09) | |
| No. of observations | 68,952 | |
| Log likelihood | -22,202 | |
| Akaike information criterion | 44,428 | |
| Bayesian information criterion | 44,538 | |

[a] *, **, *** indicates statistical significance at the 10%, 5%, and 1% level.

[b] Standard errors are shown in parenthesis.

fruit taste and texture: from a weak/mild flavor and mealy/not firm texture to a strong/full flavor and crisp/firm texture. The second-highest price premium, $0.86/lb ($1.90/kg), is for improvements in external appearance: from poor to excellent in relation to presence of defects, fruit size and green color uniformity. The third-highest premium, $0.20/lb ($0.44/kg), is for reductions in the expected number of chemical applications by 80% compared to the baseline number of applications. Also, respondents' WTP is lower by $0.36/lb ($0.79/kg) if the table grape variety is bred via CRISPR compared to conventional breeding.

The coefficient estimate for price is negative and statistically significant, implying an increase in price would, as expected, reduce utility of respondents. The alternative specific constant representing the no-purchase option (C) was positive and statistically significant, indicating that respondents preferred none (the no-purchase option) to either alternative A or B in each choice scenario. The scale heterogeneity parameter $\tau$ is statistically significant indicating substantial scale heterogeneity in the data. The estimated standard deviations are statistically significant for all the attributes, implying heterogeneity in the respondents' preferences (except for those related to the expected number of chemical applications).

The results have several practical implications. First, the estimates are consistent with previous studies in that the taste and texture attributes (e.g., sweetness, acidity, and crunchiness)

display the highest correlation with overall preferences for table grapes. This is followed by appearance-related attributes such as color [53–56]. Second, the respondents in our survey exhibit a stronger preference for attributes that are tangible, such as taste, texture and appearance. These results are aligned with previous studies that cover a range of food products. Combris et al. [57] conducted a study eliciting consumers' preferences for fresh pears and found that "taste beats food safety"; this implies that if consumers are informed about fresh produce produced with less pesticides, they would prefer the tasty alternative over the alternative that used less pesticides. This also is consistent with results from Malone and Lusk [58] who measured consumer perceptions applied to meat products (beef, pork and poultry) and found that consumers derive the highest utility from taste relative to how healthy and safe they perceive the product to be. However, one must consider that these results are dependent on the magnitude of the taste improvement and the magnitude of the improvement in healthfulness or safety.

Our results are aligned with those of Yang and Hobbs [7] and Marette, Disdier, and Beghin [36] who estimated a statistically significant discount for fresh apples developed using gene editing rather than conventional breeding. They are also consistent with results from Muringai, Fan, and Goddard [15] who found a statistically significant discount for potatoes developed using gene editing rather than conventional breeding. The next sub-section presents results from the latent class model to investigate further the sources of heterogeneity in preferences for table grapes produced using varieties developed using CRISPR.

### Latent class model results

Results from the latent class model are presented in Table 10. Recall, individuals were allocated among four groups based on the similarity of their preferences for bundles of grape attributes. Summary statistics for the four groups are presented in the table from left to right in descending order according to the level of acceptance of CRISPR. For example, group 1 exhibits a positive and statistically significant marginal utility for table grape varieties developed using

**Table 10. Parameter estimates for the latent class model to represent heterogeneity of preferences for bundles of table grape attributes[a,b].**

| Variable | Group 1 | Group 2 | Group 3 | Group 4 |
|---|---|---|---|---|
| Share of respondents in each group | 22% | 17% | 45% | 16% |
| Price | 0.07 | -0.63*** | -0.14***[a] | -0.28*** |
|  | (0.07) | (0.08) | (0.04) | (0.10) |
| Breeding technique | 0.22*** | -0.20** | -0.28*** | -2.71*** |
| Identifying and selecting desirable traits in plants and combining into one individual plant (CRISPR versus conventional breeding) | (0.07) | (0.09) | (0.04) | (0.17) |
| Fruit taste and texture e.g., sweetness, flavor, crispness, and firmness (combination of strong/ full falvor and crisp/firm texture versus combination of weak/mild flavor and mealy/and not firm firm texture) | 0.21*** (0.05) | 4.30*** (0.16) | 0.04 (0.04) | 1.45*** (0.12) |
| External appearance | 0.53*** | 2.16*** | 0.09*** | 1.25*** |
| e.g., presence of defects, fruit size, and green color uniformity (excellent versus poor) | (0.04) | (0.09) | (0.03) | (0.09) |
| Expected number of chemical applications | -0.22** | 0.79*** | 0.13*** | 0.12 |
| e.g., pesticides for insect pests and fungicides for fungal diseases (80% lower than current number of applications versus same as current number of applications) | (0.09) | (0.11) | (0.04) | (0.10) |
| Alternative specific constant—None option | -3.54*** | 2.80*** | -0.80*** | 1.79*** |
|  | (0.49) | (0.23) | (0.09) | (0.26) |

[a]*, **, *** indicates statistical significance at the 10%, 5%, and 1% level.

[b]Standard errors are shown in parentheses.

CRISPR. This group is followed by groups 2, 3, and 4 who exhibit a negative and statistically significant marginal utility for table grape varieties developed using CRISPR, and for which the magnitudes of the marginal utility range from -0.20 for group 2 to -2.71 for group 4.

The estimated coefficient for price is statistically significant and negative for groups 2, 3, and 4, but it is not statistically significant for group 1. Group 1 represents 22% of the survey respondents. Group 1 members, the ones who accept CRISPR, are not price sensitive and display a preference for the status quo in terms of the number of chemical applications. This group has a statistically significant and positive marginal utility for improvements in fruit taste and texture, and external appearance, yet these effects are smaller in magnitude compared with those for groups 2 and 4. Unlike the other three groups, this group exhibits a statistically significant and positive marginal utility for the use of CRISPR as the breeding technique.

Group 2 represents 17% of all survey respondents. This group exhibits a negative and statistically significant marginal utility for CRISPR, but this value is smaller compared to groups 3 and 4. Group 2 is identified as weakly rejecting CRISPR. The marginal utility for improvements in fruit taste and texture, external appearance, and reductions in the expected number of chemical applications, is positive and statistically significant. The marginal utilities for these three attributes are larger in magnitude compared with groups 1, 3, and 4.

Group 3 represents 45% of the survey respondents. This group exhibits a statistically significant and negative marginal utility for the use of CRISPR as the breeding technique, and the magnitude of this negative response to CRISPR is larger than that for group 2 but smaller than that for group 4; therefore, Group 3 is identified as moderately rejecting CRISPR. For this group, the coefficient estimate for improvements in fruit taste and texture is not statistically significant. This group exhibits a statistically significant and positive marginal utility for fruit external appearance, and for reductions in the expected number of chemical applications.

Group 4 represents 16% of all survey respondents. This group exhibits a statistically significant and negative marginal utility for CRISPR, and has the greatest negative response to CRISPR compared to the other groups. This group exhibits a statistically significant and positive marginal utility for improvements in taste and texture, and external appearance. The marginal utility for reducing the number of chemical applications is not statistically significant.

The summary statistics (means and standard deviations) of the variables used to identify the membership function variables and analysis of variance (ANOVA) to compare across the groups of respondents, are presented in Table 11. The sociodemographic variables include the following: male is a binary variable equaling 1 if the respondent is male; millennial is a binary variable equaling 1 if the respondent was born on or after 1981; income is a binary variable equaling 1 if the respondent reported an annual household income of $99,922/year or more ($99,922 was the average annual income across respondents); white is a binary variable equaling 1 if the respondent is of white ethnicity; education is a binary variable equaling 1 if the respondent has a bachelor's degree or higher; family size is a binary variable equaling 1 if the size of the household is three or more; number of children under 18 is a binary variable equaling 1 if the number of children under 18 in the household is one or more; frequency of consumption is a binary variable equaling 1 if the frequency of table grape consumption is greater than or equal to once per week. The ratings for the trusted sources of information on how the food purchased is produced, ratings for the level of knowledge, ratings for the perception of breeding technologies, are all on a scale of 1–5, from the lowest to the highest or most favorable rating. The sources of information were grouped according to the overall groups of interest they represented: consumer-oriented groups included activist groups and consumer's organizations; producer-oriented groups included individual farmers, farmer's organizations, food manufacturers and food retailers; government included local government and government agencies; scientific groups included medical professionals, scientific associations, and scientific

**Table 11. Comparison of membership function variables in terms of summary statistics across groups of survey respondents.**

| Variable | Mean (standard deviations shown in parentheses) | | | | Analysis of variance comparison, p-values | | | | | |
|---|---|---|---|---|---|---|---|---|---|---|
| | Group 1 | Group 2 | Group 3 | Group 4 | Group 1 vs group 2 | Group s 1 vs group 3 | Group 1 vs group 4 | Group s 2 vs group 3 | Group 2 vs group s 4 | Group s 3 vs group 4 |
| Share of respondents in each group (%) | 22 | 17 | 45 | 16 | | | | | | |
| Sociodemographics | | | | | | | | | | |
| Male (%) | 0.52 (0.50) | 0.32 (0.47) | 0.44 (0.50) | 0.27 (0.44) | 0.00 | 0.00 | 0.00 | 0.00 | 0.07 | 0.00 |
| Millennial born after 1981 (%) | 0.60 (0.49) | 0.30 (0.46) | 0.65 (0.48) | 0.26 (0.44) | 0.00 | 0.06 | 0.00 | 0.00 | 0.25 | 0.00 |
| Income $\geq$ \$99,922/year (%) | 0.47 (0.50) | 0.47 (0.50) | 0.44 (0.50) | 0.40 (0.49) | 0.85 | 0.32 | 0.02 | 0.27 | 0.02 | 0.08 |
| White (%) | 0.77 (0.42) | 0.82 (0.38) | 0.69 (0.46) | 0.81 (0.39) | 0.07 | 0.00 | 0.17 | 0.00 | 0.75 | 0.00 |
| Education $\geq$ bachelor's degree (%) | 0.61 (0.49) | 0.67 (0.47) | 0.57 (0.50) | 0.50 (0.50) | 0.04 | 0.09 | 0.00 | 0.00 | 0.00 | 0.01 |
| Family size $\geq$ 3(%) | 0.52 (0.50) | 0.56 (0.50) | 0.55 (0.50) | 0.50 (0.50) | 0.16 | 0.35 | 0.47 | 0.47 | 0.05 | 0.11 |
| No. of children under 18 $\geq$ 1(%) | 0.62 (0.49) | 0.56 (0.50) | 0.58 (0.49) | 0.58 (0.49) | 0.04 | 0.08 | 0.16 | 0.49 | 0.61 | 0.96 |
| Freq. of consumption $\geq$ once/week (%) | 0.41 (0.49) | 0.16 (0.36) | 0.41 (0.49) | 0.21 (0.40) | 0.00 | 0.97 | 0.00 | 0.00 | 0.11 | 0.00 |
| Respondents' ratings for the most trusted sources of information | | | | | | | | | | |
| Scientific groups [Medical professionals (e.g., your primary physician), scientific associations (e.g., American Association for the Advancement of Science), scientific journals (e. Nature, National Geographic)] | 4.10 (0.68) | 4.08 (0.67) | 3.74 (0.84) | 3.67 (0.86) | 0.60 | 0.00 | 0.00 | 0.00 | 0.00 | 0.08 |
| Producer oriented groups [individual farmers, farmer's organizations (e.g., California Table Grape Commission), food manufacturers (e.g., Nestle, General Mills, food retailers (e.g., Walmart, Safeway)] | 4.00 (0.72) | 3.59 (0.77) | 3.67 (0.77) | 3.50 (0.76) | 0.00 | 0.00 | 0.00 | 0.07 | 0.07 | 0.00 |
| Universities | 3.95 (0.94) | 3.84 (0.83) | 3.67 (0.10) | 3.39 (1.01) | 0.05 | 0.00 | 0.00 | 0.00 | 0.00 | 0.00 |
| Government [Local government (e.g., local mayor), government agencies (e.g., U.S. Department of Agriculture)] | 3.84 (0.86) | 3.47 (0.82) | 3.48 (0.94) | 3.15 (0.91) | 0.00 | 0.00 | 0.00 | 0.73 | 0.00 | 0.00 |
| Consumer oriented groups [activist group (e.g., Green America), consumer organization (e.g., American Council of Consumers)] | 3.77 (0.88) | 3.29 (0.82) | 3.49 (0.88) | 3.23 (0.87) | 0.00 | 0.00 | 0.00 | 0.00 | 0.30 | 0.00 |
| Social media, media (newspaper, TV, magazines), friends and family members | 3.67 (0.89) | 2.90 (0.71) | 3.40 (0.85) | 2.89 (0.73) | 0.00 | 0.00 | 0.00 | 0.00 | 0.94 | 0.00 |
| Respondents' ratings for the level of knowledge on the breeding technologies | | | | | | | | | | |
| Knowledge of genetic engineering | 3.62 (1.18) | 2.78 (1.16) | 3.40 (1.19) | 2.82 (1.25) | 0.00 | 0.00 | 0.00 | 0.00 | 0.59 | 0.00 |
| Knowledge of CRISPR | 3.39 (1.25) | 2.52 (1.15) | 3.20 (1.21) | 2.50 (1.17) | 0.00 | 0.00 | 0.00 | 0.00 | 0.74 | 0.00 |
| Respondents' ratings on the perception of breeding technologies | | | | | | | | | | |

(*Continued*)

**Table 11.** (Continued)

| Variable | Mean (standard deviations shown in parentheses) | | | | Analysis of variance comparison, p-values | | | | | |
|---|---|---|---|---|---|---|---|---|---|---|
| | Group 1 | Group 2 | Group 3 | Group 4 | Group 1 vs. group 2 | Group s 1 vs. group 3 | Group 1 vs. group 4 | Group s 2 vs. group 3 | Group 2 vs. group s 4 | Group s 3 vs. group 4 |
| CRISPR is safe | 3.67 (1.07) | 3.56 (1.01) | 3.27 (1.07) | 2.52 (0.97) | 0.06 | 0.00 | 0.00 | 0.00 | 0.00 | 0.00 |
| Genetic engineering is safe | 3.62 (1.14) | 3.46 (1.12) | 3.20 (1.19) | 2.30 (1.06) | 0.02 | 0.00 | 0.00 | 0.00 | 0.00 | 0.00 |
| CRISPR is natural | 3.29 (1.23) | 2.57 (1.07) | 3.07 (1.23) | 2.02 (0.97) | 0.00 | 0.00 | 0.00 | 0.00 | 0.00 | 0.00 |
| Genetic engineering is natural | 3.16 (1.29) | 2.43 (1.11) | 2.98 (1.29) | 1.79 (0.95) | 0.00 | 0.00 | 0.00 | 0.00 | 0.00 | 0.00 |

journals; social media, media, friends and family were grouped together; and universities were grouped separately from government agency and scientific group. The ratings for each source were averaged across the group of interest, and one single rating for the group of interest was included in the model.

Group 1 is the group of respondents that prefer CRISPR over conventional breeding. Across all four groups, group 1 has the largest proportion of males and the largest proportion of families with children under 18. Along with group 2, group 1 has the largest proportion of respondents with an income higher than $99,922/year. Group 1 and group 3 include larger proportions of individuals who consume table grapes more often than once a week. Across all four groups, group 1 and group 2 trust scientific organizations the most. Across all four groups, group 1 is the most trusting of the different sources of information, including producer-oriented organizations, universities, government, consumer-oriented organizations, and social media, media, friends and family. Across all four groups, group 1 self reports that they have the most knowledge about genetic engineering and CRISPR. Also across all four groups, group 1 more strongly perceives both genetic engineering and CRISPR as safe, natural, ethical and morally acceptable.

Across all four groups, group 4 most strongly rejects CRISPR. This group has a greater proportion of females and the smallest proportion of millennials (born after 1981). Group 4 also has the smallest proportion of individuals with a bachelor's degree or higher. Across all four groups, group 4 distrusts scientific organizations, producer-oriented organizations, universities, and government. In terms of knowledge, group 4 (and group 2) self-reported knowing the least about CRISPR across all four groups. Also, group 4 most strongly perceives CRISPR and genetic engineering as not safe, not natural and not ethical and morally acceptable.

In brief, in terms of the acceptance of CRISPR, we have identified two contrasting sets of groups: group 1 favors the use of CRISPR versus groups, 2, 3, 4 that reject it. The nature of the rejection varies among the groups: group 2 weakly rejects CRISPR, group 3 moderately rejects CRISPR, and group 4 strongly rejects CRISPR. Group 1 self-reports knowing more about genetic engineering and CRISPR and, compared with the other groups, considers both breeding technologies to be safe, natural and ethical and morally acceptable.

## Summary and conclusion

The implementation of new plant breeding technologies, and in particular genetic engineering, has been controversial. Despite the consensus in the scientific community that these

technologies are safe and effective, many consumers prefer to avoid genetically engineered food products and many food processors, manufacturers, and retailers seek to avoid using or selling these products. This resistance in the marketplace has prevented realizing the benefits of these new technologies to their full potential.

This study estimates the willingness to pay for bundles of genetic traits and other attributes of table grapes, including fruit taste and texture, external appearance, expected number of chemical applications, and the breeding technology used to develop the table grape varieties (conventional breeding versus CRISPR). This is an interesting and relevant case study because table grapes are a fresh fruit product from an industry that has witnessed significant varietal development over the past 30 years, and is very actively developing and introducing new culti-vars; it is an industry in which the demand for varietal innovation is clearly strong.

Previous studies have shown evidence that part of the consumer resistance to genetically engineered food products stems from a lack of awareness of the benefits associated with the genetically engineered varieties. The benefits from the main genetically engineered crop varie-ties are related to increased productivity, and resistance to diseases and insect pests, and these benefits which lead to lower production costs and subsequently to lower food prices, are not apparent as such to final consumers. Unlike most previous studies, in this study we consider bundles of attributes that imply a direct tangible benefit to consumers, such as fruit taste and texture and external appearance, as well as bundles that imply a direct benefit to producers (and indirect non-tangible benefit to consumers) such as a reduced number of chemical applications.

Results from this study suggest that respondents as consumers are willing to pay the highest price premiums for improvements in table grape taste and texture, followed by improvements in external appearance, and then reductions in the expected number of chemical applications. Also, most of the respondents would apply a discount to table grapes produced from varieties developed via CRISPR compared to those produced using conventional breeding.

Results from a latent class model identify four distinct consumer groups. These groups dif-fer in their marginal utility derived from improvements in fruit taste and texture, from reduc-tions in the expected number of chemical applications, and from the breeding technique used. Compared with the other three groups, members of the group that favors gene-editing (group 1) are more likely to be males, and to have more than one child in the household. This group self-reported knowing more about genetic engineering compared to the other groups that reject CRISPR. Also, the group that strongly rejects CRISPR (group 4) considers both genetic engineering and CRISPR to be breeding methods that produce foods that are not safe to eat, and CRISPR as a breeding method that is not ethical or morally acceptable.

Several real-world implications stem from our results. First, like previous studies, respon-dents to our survey favor fruit taste and texture over other groups of attributes. Second, based on the stated discounts for CRISPR estimated in this study, and the results from the latent class model, one expects that CRISPR and gene editing in general would face some barriers in the marketplace, as some resistance to these technologies among consumers may remain. Third, this study provides evidence about which sources of information are trusted by each group, revealing that the group that accepts gene editing (group 1) is—in general—most trust-ful of information coming from scientific organizations and social media sources, media, and friends and family. The group that most strongly rejects gene editing (group 4) consistently distrusts information from all sources. Further research that aims to improve our understand-ing of the effect of various sources of information on consumers' acceptance of new plant breeding technologies is warranted.

## Supporting information

**S1 File.**
(ZIP)

## Author Contributions

**Conceptualization:** R. Karina Gallardo, Bradley Rickard, Julian Alston.

**Data curation:** R. Karina Gallardo, Bradley Rickard, Julian Alston.

**Formal analysis:** Azhar Uddin, R. Karina Gallardo, Bradley Rickard, Julian Alston.

**Funding acquisition:** R. Karina Gallardo, Bradley Rickard, Julian Alston.

**Investigation:** R. Karina Gallardo, Bradley Rickard, Julian Alston.

**Methodology:** R. Karina Gallardo, Bradley Rickard, Julian Alston, Olena Sambucci.

**Project administration:** R. Karina Gallardo, Julian Alston.

**Resources:** R. Karina Gallardo, Julian Alston.

**Software:** Azhar Uddin.

**Supervision:** R. Karina Gallardo, Bradley Rickard, Olena Sambucci.

**Validation:** R. Karina Gallardo, Bradley Rickard, Julian Alston.

**Visualization:** R. Karina Gallardo, Bradley Rickard, Julian Alston.

**Writing – original draft:** Azhar Uddin, R. Karina Gallardo, Bradley Rickard, Julian Alston, Olena Sambucci.

**Writing – review & editing:** R. Karina Gallardo, Bradley Rickard, Julian Alston.

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
