## [Decision Letter · Decision Letter 0]

25 Mar 2022

PONE-D-22-03341Consumer Acceptance of New Plant-Breeding Technologies: An Application to the Use of Gene Editing in Fresh Table GrapesPLOS ONE

Dear Dr. Gallardo,

Thank you for submitting your manuscript to PLOS ONE. After careful consideration, we feel that it has merit but does not fully meet PLOS ONE’s publication criteria as it currently stands. Therefore, we invite you to submit a revised version of the manuscript that addresses the points raised during the review process.

 Reviewer #2 have provided important comments to improve this manuscript. I would like to ask you to response to those comments properly.

We look forward to receiving your revised manuscript.

Kind regards,

Hiroshi Ezura

Academic Editor

PLOS ONE

Journal Requirements:

[This work is supported in part by the USDA National Institute of Food and Agriculture - Specialty Crop Research Initiative project “VitisGEN2” (2017-51181-26829).]

[RKG, BR, JA received funding

USDA National Institute of Food and Agriculture - Specialty Crop Research Initiative project “VitisGEN2” (2017-51181-26829)

https://www.vitisgen2.org/

https://nifa.usda.gov/funding-opportunity/specialty-crop-research-initiative-scri

NO.

The funders had no role in study design, data collection and analysis, decision to publish, or preparation of the manuscript.]

6. We note that you have referenced (ie. Bewick et al. [5]) which has currently not yet been accepted for publication. Please remove this from your References and amend this to state in the body of your manuscript: (ie “Bewick et al. [Unpublished]”) as detailed online in our guide for authors

http://journals.plos.org/plosone/s/submission-guidelines#loc-reference-style.

Reviewers' comments:

Reviewer's Responses to Questions

**Comments to the Author**

1. Is the manuscript technically sound, and do the data support the conclusions?

Reviewer #1: Yes

Reviewer #2: Partly

2. Has the statistical analysis been performed appropriately and rigorously? 

Reviewer #1: Yes

Reviewer #2: I Don't Know

3. Have the authors made all data underlying the findings in their manuscript fully available?

Reviewer #1: Yes

Reviewer #2: Yes

4. Is the manuscript presented in an intelligible fashion and written in standard English?

Reviewer #1: Yes

Reviewer #2: Yes

5. Review Comments to the Author

Reviewer #1: In this manuscript entitled "Consumer Acceptance of New Plant-Breeding Technologies: An Application to the Use of Gene Editing in Fresh Table Grapes", the authors estimate the willingness to pay for genetic traits and other attributes of table grape in comparison with the conventional breeding method and the gene editing.

This study on the estimation of willingness to pay of a large number of more than 2,800 consumers based on a wide decision branches in breeding techniques, agricultural management, and product value in biotechnology foods developed with fruit trees that take time to breed will provide important light on for future development goal of biotech plants using genome editing and for planning the better way to market them.

The investigation and statistical analysis are well performed, and the results are convincing.

I have a few relatively minor comments, explained below.

-The first letter of both Genetically Engineering and Gene Editing is "GE". Although there is a definition in the manuscript, it may cause confusion to reader, so could the author change the abbreviation to something more clear?

-The term "Fig" should have a period in it. Please consider this across the manuscript.

-For example on Page 4, line 5 in Background section, since hyphens are used in other CRISPR, "CRISPR/Cas9" should be replaced with a hyphen. Please consider this across the manuscript.

-In the characteristics of the survey in Table 3 on page 14, there seems to be a tendency for those in "Education" to be slightly more educated and those in "Income" to have higher income. It may expected that this tendency has some influence on the acceptance of biotech foods. What is the author's opinion on this point? You don't necessarily have to answer.

-In the Table 4 on page 16, a total of 9.3% of the respondents have a consumption frequency of 2-3 times a year or less than 2-3 times a year. How do these consumers conduct their consumption activities?

-The hindmost sentence of the first paragraph on the page 27 has no period.

Reviewer #2: This paper analyses consumers’ preferences with regard to fresh produce by comparing gene-edited table grapes and grapes bred by conventional breeding methods. Using consumers’ willingness to pay for attributes of table grapes, especially paying attention to attributes of quality such as taste and texture and agronomic characteristics such as the number of chemical applications, the paper identifies hierarchies among the identified attributes and substantiates them with empirical data. The justification of why fresh table grapes were chosen for this study is logically sound and persuasive; thus readers who are interested in carrying out similar studies using other kinds of produce can see to what extent and how the proposed analytical framework can be used for their studies.

The main contribution of the paper is to advance knowledge about consumer acceptance of gene-edited foods. The topic is important and timely considering the fact that gene-edited foods are beginning to be introduced to the market in some parts of the world. In addition, I would like to add that the significance of this study is that the paper has succeeded in analytically and empirically unbundling the relevant attributes of gene-edited table grapes, which previously were bundled together as simply produce that has merits for producers or consumers.

Unfortunately, my expertise does not allow me to evaluate the statical analysis performed for this paper, thus I defer on this to the other reviewers who have the relevant background.

I have chosen “yes” for most of the review questions in the Plos One editorial manager but have chosen “partly” for the question that asks “Is the manuscript technically sound, and do the data support the conclusions?” Given that, I recommend that this paper be accepted for publication with major revision.

The reason I have chosen “partly” for the technical soundness of this manuscript pertains to the ways in which the review of literature is organized. While the review concerning the empirical approach is elaborate, reviews of the key concepts are somewhat insufficient. I see that some of the fundamental and recent papers in the field that examines consumer acceptance, or more broadly public acceptance, of breeding technologies, are not included in this paper, though the authors suggest that the main contribution of the paper is to advance the knowledge about consumer acceptance of new plant-breeding technologies, building on what has been done previously. I agree that studies of consumer acceptance of gene-edited foods are an emerging field of study, and thus scholarship on this particular topic is not as solid as for other fields of study such as consumer acceptance of genetically modified foods, cisgenics, mutagenesis, or organic thereof, but these existing studies are quite relevant and useful for this paper.

Depending on how the authors want to organize this section, the authors could expand on the fields of application of gene editing technologies to animals, as studies of public views on gene edited animals are quite extensive and could again provide useful insights for the present paper. Alternatively, the paper can bring in literature that sheds light on citizens', the public's or stakeholders’ views of gene edited crops and foods; this should help readers to situate the topic of consumer acceptance of gene-edited foods within the broader literature.

Additionally, I would like to point out that some of the interesting points raised in the “background” section (from third line, page 6 onwards) that begins “Our research is motivated by questions regarding the …” could be moved to the next section (“Consumer response to food produced with gene editing”) so as to add additional dimensions to the discussion of consumer response. In short, you might want to consider reorganizing the earlier sections of the paper so that the research questions for this paper are clearly laid out before the paper walks readers through its empirical approach.

Lack of clarification of the key concepts in this paper seems to have had some ramifications for the discussion in the concluding section.

I was slightly confused by some of the points raised in the conclusion. For instance, the first sentence of paragraph three introduces a point that alludes to a relationship between consumer resistance and awareness of benefits associated with GE, but it is not clear where this point in the conclusion stemmed from in terms of the earlier presentation of the data or in terms of its relationship to the existing literature. Another example would be the point raised in the final paragraph of the concluding section that says “Second, gene editing is expected to face some barriers in the marketplace, emphasizing the idea that main opponents will remain opposing these technologies.” I might’ve missed it, but it is not clear why the authors expect barriers to be present in the marketplace and who the main opponents are.

Finally, correction or further clarification is required in these specific spots:

1. Page 5, Line 7~8: “…the use of GE breeding techniques in humans and animals is of concern for some groups…”

2. Page 6, Line 21~22: “A version to GE foods varies among products and across consumers, and this variation depends largely on the direct benefit perceived by consumers, their…” It is not clear what “a version” means in this instance.

3. Page 28, Line 11: “The list of food products included fresh table grapes, fresh milk and raw potatoes, and also processed versions of each product (namely grape juice, ice cream, and French fries).” It is not clear why grapes are compared with milk and potatoes.

In summary, although the earlier section of the paper needs to be reorganized, I believe that the authors should be able to address these appropriately with modifications to the text.

6. PLOS authors have the option to publish the peer review history of their article (what does this mean?). If published, this will include your full peer review and any attached files.

Reviewer #1: No

Reviewer #2: **Yes**

---

## [Author Response · Author response to Decision Letter 0]

12 May 2022

Response to Reviewers

PONE-D-22-03341

Consumer Acceptance of New Plant-Breeding Technologies: An Application to the Use of Gene Editing in Fresh Table Grapes

https://journals.plos.org/plosone/s/file?id=wjVg/PLOSOne_formatting_sample_main_body.pdfand

Response: Thank you for the reminder, we have ensured to the best of our capabilities that the manuscript and the title page meet PLOS ONE style requirements. 

Response: We include the title page at the beginning of the manuscript file itself, with the listing of all authors and affiliations.

[This work is supported in part by the USDA National Institute of Food and Agriculture - Specialty Crop Research Initiative project “VitisGEN2” (2017-51181-26829).]

[RKG, BR, JA, AU, OS received funding

This work is supported in part by the USDA National Institute of Food and Agriculture - Specialty Crop Research Initiative project “VitisGEN2” (2017-51181-26829)

https://www.vitisgen2.org/

https://nifa.usda.gov/funding-opportunity/specialty-crop-research-initiative-scri

NO.

The funders had no role in study design, data collection and analysis, decision to publish, or preparation of the manuscript.]

Response: We include the amended statements within the cover letter. Thank you for changing the online submission form on our behalf.

Response: In the cover letter we address the prompt that we acknowledge there are no restrictions on sharing a de-identified data set. Therefore, we are uploading a minimal anonymized data set necessary to replicate this study findings, as Supporting Information Files.

Response: The full ethics statement is now included in the Data section: The survey tool was approved by the Washington State University Institutional Review Board (IRB) for the use of human subjects. The IRB number is 18186-001.

6. We note that you have referenced (ie. Bewick et al. [5]) which has currently not yet been accepted for publication. Please remove this from your References and amend this to state in the body of your manuscript: (ie “Bewick et al. [Unpublished]”) as detailed online in our guide for authors

Response: We have checked the manuscript several times and are not able to find the Bewick reference. We noticed that the reference to Alston, Sambucci, and Serfas is unpublished and removed from the references and amended in the body of the manuscript as Alston, Sambucci, and Serfas [Unpublished]. 

Reviewer #1

 In this manuscript entitled "Consumer Acceptance of New Plant-Breeding Technologies: An Application to the Use of Gene Editing in Fresh Table Grapes", the authors estimate the willingness to pay for genetic traits and other attributes of table grape in comparison with the conventional breeding method and the gene editing.

This study on the estimation of willingness to pay of a large number of more than 2,800 consumers based on a wide decision branches in breeding techniques, agricultural management, and product value in biotechnology foods developed with fruit trees that take time to breed will provide important light on for future development goal of biotech plants using genome editing and for planning the better way to market them.

The investigation and statistical analysis are well performed, and the results are convincing.

I have a few relatively minor comments, explained below.

-The first letter of both Genetically Engineering and Gene Editing is "GE". Although there is a definition in the manuscript, it may cause confusion to reader, so could the author change the abbreviation to something more clear?

Response: Thank you for the suggestion. The revised version of the manuscript does not use the abbreviation GE, because we agree that its use could induce confusion. We use the term genetic engineering instead of using GE. Because we explicitly indicated in the survey that gene editing referred to CRISPR, we use CRISPR to refer to gene editing in the description of the study and discussion of results.

-The term “Fig” should have a period in it. Please consider this across the manuscript.

Response: Thank you for the suggestion. However, please know that in the PLOS ONE style templates there is no period after Fig, see this link:

https://journals.plos.org/plosone/s/file?id=wjVg/PLOSOne_formatting_sample_main_body.pdfa

-For example on Page 4, line 5 in Background section, since hyphens are used in other CRISPR, “CRISPR/Cas9” should be replaced with a hyphen. Please consider this across the manuscript.

Response: Thank you for the observation, we now using CRISPR to refer to CRISPR-Cas9, we don’t use CRISPR-Cas9 in the manuscript. We added a sentence to clarify: CRISPR-Cas systems, referred in this manuscript as CRISPR.

-In the characteristics of the survey in Table 3 on page 14, there seems to be a tendency for those in “Education” to be slightly more educated and those in “Income” to have higher income. It may expected that this tendency has some influence on the acceptance of biotech foods. What is the author’s opinion on this point? You don’t necessarily have to answer.

Response: Thank you for the comment. You are absolutely correct to point out that our sample is not representative of education and income of the general U.S. population. Therefore, we present all the results, not mentioning they are applicable to consumers at large but applicable to respondents in our survey sample, so the reader is aware that our results apply to our sample and may not generalize to the U.S. population. The revised manuscript now says this explicitly when presenting the WTP results:

Table 9 presents the WTP results from the GMNL-I model; recall these results are applicable to the respondents to the survey in this study.

-In the Table 4 on page 16, a total of 9.3% of the respondents have a consumption frequency of 2-3 times a year or less than 2-3 times a year. How do these consumers conduct their consumption activities?

Response: We did not include in the survey a question asking how consumers conducted their consumption activities. We asked about frequency of consumption, the color of preferred grapes (black, red, green), and if the type of preferred package loosed, pre-bagged or clamshell. 

-The hindmost sentence of the first paragraph on the page 27 has no period.

Response: Thank you for catching this typo. We added the period to the sentence.

Unlike the other three groups, this group exhibits a statistically significant and positive marginal utility for the use CRISPR as the breeding technique. 

Reviewer #2

This paper analyses consumers’ preferences with regard to fresh produce by comparing gene-edited table grapes and grapes bred by conventional breeding methods. Using consumers’ willingness to pay for attributes of table grapes, especially paying attention to attributes of quality such as taste and texture and agronomic characteristics such as the number of chemical applications, the paper identifies hierarchies among the identified attributes and substantiates them with empirical data. The justification of why fresh table grapes were chosen for this study is logically sound and persuasive; thus readers who are interested in carrying out similar studies using other kinds of produce can see to what extent and how the proposed analytical framework can be used for their studies.

The main contribution of the paper is to advance knowledge about consumer acceptance of gene-edited foods. The topic is important and timely considering the fact that gene-edited foods are beginning to be introduced to the market in some parts of the world. In addition, I would like to add that the significance of this study is that the paper has succeeded in analytically and empirically unbundling the relevant attributes of gene-edited table grapes, which previously were bundled together as simply produce that has merits for producers or consumers.

Unfortunately, my expertise does not allow me to evaluate the statical analysis performed for this paper, thus I defer on this to the other reviewers who have the relevant background.

I have chosen “yes” for most of the review questions in the Plos One editorial manager but have chosen “partly” for the question that asks “Is the manuscript technically sound, and do the data support the conclusions?” Given that, I recommend that this paper be accepted for publication with major revision.

The reason I have chosen “partly” for the technical soundness of this manuscript pertains to the ways in which the review of literature is organized. While the review concerning the empirical approach is elaborate, reviews of the key concepts are somewhat insufficient. I see that some of the fundamental and recent papers in the field that examines consumer acceptance, or more broadly public acceptance, of breeding technologies, are not included in this paper, though the authors suggest that the main contribution of the paper is to advance the knowledge about consumer acceptance of new plant-breeding technologies, building on what has been done previously. I agree that studies of consumer acceptance of gene-edited foods are an emerging field of study, and thus scholarship on this particular topic is not as solid as for other fields of study such as consumer acceptance of genetically modified foods, cisgenics, mutagenesis, or organic thereof, but these existing studies are quite relevant and useful for this paper.

Depending on how the authors want to organize this section, the authors could expand on the fields of application of gene editing technologies to animals, as studies of public views on gene edited animals are quite extensive and could again provide useful insights for the present paper. 

Alternatively, the paper can bring in literature that sheds light on citizens', the public's or stakeholders’ views of gene edited crops and foods; this should help readers to situate the topic of consumer acceptance of gene-edited foods within the broader literature.

Response: Thank you for this suggestion. We have introduced the following modifications to the manuscript. These details could be revised to reflect subsequent edits made to the quoted parts of the paper. 

• We added to the literature review narrative that discusses applications of gene editing in animal production: 

The present study focuses on CRIPSR applications to plant crops. However, we expand on the fields of application of gene editing to animals. In general, CRISPR system technologies have been used to treat genetic disorders in animals, expedite livestock breeding, and to engineer new antimicrobials and control disease carrying insects with gene [23]. If successfully applied, gene editing can improve animal productive traits, disease-resistance in animals, making livestock more efficient, environmentally sustainable, and improving animal welfare [24]. Examples of the applications include the Superfine Merino lambs that produce the highest quality of wool and improved meat production, and pigs resistant to the Porcine Respiratory and Reproductive Syndrome virus. In terms of animal welfare, gene editing can be used to avoid practices such as horn removal in calves [24]. 

We also added a review of the literature that focuses on citizens', the public's or stakeholders’ views of gene edited crops and foods: 

The applications of gene editing go beyond improvements in agricultural crops, and can be applied to any living organism, including animals and humans [21]. Results from different studies show that, in general, the use of gene editing in humans and animals is of concern for some individuals, notwithstanding a very broad scientific consensus as to their safety for the environment and human health [21, 28, 29]. For example, Critchley et al. [21] showed that individuals could be more supportive if the result is enhanced human health, but no support is shown to applications on animals for human food and enhancement of human appearance. Watanabe et al. [28] underscores the increasing controversy surrounding the application of gene editing to humans, either for therapeutic or function enhancement. Funk et al. [29] showed that Americans are concerned about using gene editing to reduce the risk of diseases in humans. 

Additionally, I would like to point out that some of the interesting points raised in the “background” section (from third line, page 6 onwards) that begins “Our research is motivated by questions regarding the …” could be moved to the next section (“Consumer response to food produced with gene editing”) so as to add additional dimensions to the discussion of consumer response. In short, you might want to consider reorganizing the earlier sections of the paper so that the research questions for this paper are clearly laid out before the paper walks readers through its empirical approach.

Response: We have moved the paragraph that starts with “Our research is motivated by questions regarding the …” under “Consumer response to food produced with gene editing”. We also moved the subsequent paragraphs to the paragraph mentioned, right after, in order to follow the logical sequence of the ideas presented.

Lack of clarification of the key concepts in this paper seems to have had some ramifications for the discussion in the concluding section.

I was slightly confused by some of the points raised in the conclusion. For instance, the first sentence of paragraph three introduces a point that alludes to a relationship between consumer resistance and awareness of benefits associated with GE, but it is not clear where this point in the conclusion stemmed from in terms of the earlier presentation of the data or in terms of its relationship to the existing literature. 

Response: Thank you for this comment. We have clarified this issue in the concluding section. These details might have to be revised to reflect subsequent edits made to the quoted parts of the paper.

Existing literature has shown evidence that part of the consumer resistance to genetically engineering food products stems from lack of awareness of the benefits associated with the genetically engineering varieties. Also, existing literature concludes that the benefits from the main genetically engineering crop varieties are related to increased productivity, resistance to diseases and insect pests. These benefits manifest into lower producer costs that are not apparent to final consumers though they may be reflected indirectly in lower food prices. Different from the existing literature, in this study we include bundles of attributes that imply a direct tangible benefit to consumers, such as fruit taste and texture and external appearance, as well as bundles that imply a direct benefit to producers (and indirect non-tangible benefit to consumers) such as a reduced number of chemical applications.

Another example would be the point raised in the final paragraph of the concluding section that says “Second, gene editing is expected to face some barriers in the marketplace, emphasizing the idea that main opponents will remain opposing these technologies.” I might’ve missed it, but it is not clear why the authors expect barriers to be present in the marketplace and who the main opponents are.

Response: Thank you for the suggestion. These details might have to be revised to reflect subsequent edits made to the quoted parts of the paper.

Second, based on the stated discounts estimated in this study for CRISPR, and the results from the latent class model, one expects that CRISPR and in general gene editing would face some barriers in the marketplace, as some resistance to these technologies among consumers may remain.

Finally, correction or further clarification is required in these specific spots:

1. Page 5, Line 7~8: “…the use of GE breeding techniques in humans and animals is of concern for some groups…”

Response: The line has been modified: “Results from different studies show that in general the use of gene editing in humans and animals is of concern for some individuals, …”

2. Page 6, Line 21~22: “A version to GE foods varies among products and across consumers, and this variation depends largely on the direct benefit perceived by consumers, their…” It is not clear what “a version” means in this instance.

Response: This was a typo. The sentence now reads: “The aversion to genetically engineering foods …”

3. Page 28, Line 11: “The list of food products included fresh table grapes, fresh milk and raw potatoes, and also processed versions of each product (namely grape juice, ice cream, and French fries).” It is not clear why grapes are compared with milk and potatoes. 

Response: We added the explanation of why milk and potatoes were compared to fresh grapes. These details might have to be revised to reflect subsequent edits made to the quoted parts of the paper: Milk and potatoes were included as examples of foods—unlike fresh fruits—that could be consumed both fresh and processed, where grapes, milk and potatoes are the main ingredient. 

In summary, although the earlier section of the paper needs to be reorganized, I believe that the authors should be able to address these appropriately with modifications to the text.

Response: Thank you for the suggestion. The earlier sections of the paper have been reorganized following your suggestions.

---

## [Decision Letter · Decision Letter 1]

20 Jun 2022

Consumer Acceptance of New Plant-Breeding Technologies: An Application to the Use of Gene Editing in Fresh Table Grapes

PONE-D-22-03341R1

Dear Dr. Gallardo,

We’re pleased to inform you that your manuscript has been judged scientifically suitable for publication and will be formally accepted for publication once it meets all outstanding technical requirements.

Kind regards,

Hiroshi Ezura

Academic Editor

PLOS ONE

Additional Editor Comments (optional):

Reviewers' comments:

Reviewer's Responses to Questions

**Comments to the Author**

1. If the authors have adequately addressed your comments raised in a previous round of review and you feel that this manuscript is now acceptable for publication, you may indicate that here to bypass the “Comments to the Author” section, enter your conflict of interest statement in the “Confidential to Editor” section, and submit your "Accept" recommendation.

Reviewer #1: All comments have been addressed

Reviewer #2: All comments have been addressed

2. Is the manuscript technically sound, and do the data support the conclusions?

Reviewer #1: Yes

Reviewer #2: Yes

3. Has the statistical analysis been performed appropriately and rigorously? 

Reviewer #1: Yes

Reviewer #2: I Don't Know

4. Have the authors made all data underlying the findings in their manuscript fully available?

Reviewer #1: Yes

Reviewer #2: Yes

5. Is the manuscript presented in an intelligible fashion and written in standard English?

Reviewer #1: Yes

Reviewer #2: Yes

6. Review Comments to the Author

Reviewer #1: (No Response)

Reviewer #2: All of suggestions have been appropirately incorpoted in the revised manuscript. If I may, I would like to suggest authors to add a sentence or two (page 6, second paragraph) to explain why authors think it is useful to build your study on the studies of gene editring animals, while the theme of paper is "consumer acceptance of plants gene editing". Demonstrating how two bodies of literature connect to one another should stregthen your argument.

7. PLOS authors have the option to publish the peer review history of their article (what does this mean?). If published, this will include your full peer review and any attached files.

Reviewer #1: No

Reviewer #2: No

---

## [Editor Report · Acceptance letter]

27 Jul 2022

PONE-D-22-03341R1 

Consumer Acceptance of New Plant-Breeding Technologies: An Application to the Use of Gene Editing in Fresh Table Grapes 

Dear Dr. Gallardo:

I'm pleased to inform you that your manuscript has been deemed suitable for publication in PLOS ONE. Congratulations! Your manuscript is now with our production department. 

Kind regards, 

on behalf of

Prof. Hiroshi Ezura 

Academic Editor

PLOS ONE